

# Urban–rural differences in perception of trees described by parents bringing up children in Warsaw and Jedlińsk, Poland

Magdalena Wojnowska-Heciak[1,*], Magdalena Błaszczyk[2],
Marzena Suchocka[2] and Joanna Kosno-Jończy[3,*]

[1] Department of Civil Engineering and Architecture, Kielce University of Technology, Kielce, Poland
[2] Department of Landscape Architecture, Institute of Environmental Engineering, Warsaw University of Life Sciences-SGGW, Warsaw, Poland
[3] Unaffiliated, Warsaw, Poland
* These authors contributed equally to this work.

## ABSTRACT

Parents' attitudes to trees and nature are reflected not only in their children's outdoor activity, but also in the way they perceive, learn and value the environment. One hundred and eleven respondents, divided into two groups by place of residence, assessed statements in a survey questionnaire. Two groups of questions aimed at evaluating tree benefits and disservices as perceived by urban and rural parents, and identifying their preferences concerning outdoor activity of their children. Tree benefits and disadvantages were grouped into five categories (social, economic, environmental, health and aesthetic). Both urban and rural parents presented similar attitudes to trees as well as to their children's play environments. Among 37 statements concerning tree benefits, only five revealed statistically significant differences. The most important difference appeared in the way urban and rural parents perceived the aspects of danger. Trees were not perceived as posing any risk on playgrounds for city residents, who—unlike villagers—opposed to the removal of trees from playgrounds.

## INTRODUCTION

The issue of children's development in today's booming world is complicated, especially in the context of the role of nature and trees for the proper development of children considering the reduced amount of time spent outdoor by the children nowadays. The study requires an analysis of the perception of services and disservices provided by trees in cities and in rural areas as perceived by adults who control children' timetables. Another issue is how the contact with nature influences children's development, especially in the context of differences between the current and previous generations. All these aspects, described in the literature review below, allowed us to build a coherent structure of our research.

Corresponding author
Magdalena Wojnowska-Heciak,
mwojnowska@tu.kielce.pl

Perception is the basic process of cognition. The ability to observe and receive stimuli from the environment enables people to function in it and allows them to get to know each other. The main channels through which raw information (sensations) reaches the human brain are the senses, of which sight, hearing and touch play the greatest role in spatial orientation. Vision (that is, interpreted sensation) is a complex entity and best adapted to the perception of space of all senses (*Hatfield, 2001*). In addition to spatial impressions, vision receives the time-related experience (*Klincewicz, 2014*) and provides the ability to assess not only the physical properties of space (shapes, colours, light-shade systems), but also movement and change. Terms such as "perception", "sensual impression", "experience" and "cognition" describe ways or levels of learning and understanding space (*Bernaciak, 2014*).

Trees play a huge role in the perception of space. They affect the senses both as a single element and a part of the landscape. Trees can be perceived in a variety of ways, depending on the place of birth, education, upbringing or views of the observer. For some people, they are a part of social and mental life. Others, however, are negatively minded, noticing mostly defects and threats from trees. Numerous studies have demonstrated that the presence of trees and urban nature can improve people's mental and physical health, children's attention and test scores, the property values in a neighbourhood and beyond. Trees cool our urban centres. Trees are essential for healthy communities and people (*Turner-Skoff & Cavender, 2019*).

## Perception of tree benefits and threats

Positive perception of trees and the benefits of their presence in a built environment (*Braverman, 2008*) are verified in various types of surveys. This is reflected in the respondents' indications of a greater sense of security (*Rae, Simon & Braden, 2010*; *Brunson, 1999*; *Kuo, Bacaicoa & Sullivan, 1998*; *Schroeder & Anderson, 1985*; *Shaffer & Anderson, 1985*; *Kuo & Sullivan, 2001*), energy savings for the building's cooling (*Heisler, 1986*; *Kaplan, 1987*; *Kirkpatrick, Davison & Daniels, 2012*; *Simpson, 2002*; *Akbari, Pomerantz & Taha, 2001*; *Nowak & Dwyer, 2007*; *Escobedo, Kroeger & Wagner, 2011*), improved microclimate of the city and mitigated heat island effect (*Peper et al., 2010*; *McPherson, 1993*; *Potchter, Cohen & Bitan, 2006*; *Bowler et al., 2010*; *Shashua-Bar et al., 2009*; *McCarthy, Best & Betts, 2010*), along with reduced amount of carbon dioxide (*Nowak & Dwyer, 2007*; *Escobedo, Kroeger & Wagner, 2011*) rainfall runoff, noise levels and wind speed (*Nowak & Dwyer, 2007*; *Tyrväinen et al., 2005*). People associate relief from stress with being surrounded by nature (*Jiang, Chang & Sullivan, 2014*; *Jiang et al., 2016*; *Van den Berg, Jorgensen & Wilson, 2014*; *Heinrichs et al., 2003*). The literature on environmental preference and restoration is guided also by Stress Recovery Theory from Ulrich (SRT) (*Ulrich, 1983*; *Ulrich et al., 1991*) and Attention Restoration Theory (ART) from *Kaplan & Kaplan (1989)*. Visual and aesthetic values of trees are also highly appreciated (*Sudipto & Pickering, 2012*). Economic benefits of trees are well known, too. The presence of trees (especially big and mature ones) contributes to the increase of property value (*Ames & Dewald, 2003*; *Wolch, Byrne & Newell, 2014*). Social and psychological benefits associated with trees and nature are also well recognised. A stronger

sense of community and connexion with other people, as well as increasing feeling of safety and enjoyment of everyday life, is reported (*Tyrväinen et al., 2005*; *Bhatti & Church, 2004*; *Nilsson et al., 2011*; *De Vries et al., 2003*; *Taylor et al., 2015*; *Korpela et al., 2014*; *Nilsson et al., 2018*; *Kondo et al., 2018*).

Although tree benefits seem to be overwhelming (*Suchocka, Jankowski & Błaszczyk, 2019a*), some tree related annoyance is also reported. Displeasure with messiness and clutter, nuisance caused by insects and animals, and allergies are among the reported inconveniences (*Dwyer et al., 1992*). Complaints concern damage caused by tree roots lifting sidewalks and clogging underground pipes (*Rolf & Stal, 1994*; *Östberg et al., 2012*). Trees can also trigger a sense of danger in case of defective branches or poor tree architecture. As the level of acceptable risk is rather low, trees are often felled, as a result of an unsubstantiated fear. Trees' conflict with the road infrastructure is one of the main reasons of felling trees in Poland, and law regulations seem to be the most efficient mean of tree protection (*Suchocka et al., 1816*; *Suchocka, Jankowski & Błaszczyk, 2019b*). In general, trees are perceived as a threat to traffic safety, especially by road authorities. Trees 'killing the drivers' is a common picture presented to the public, also by the media. Tall, mature trees that block off light can also lead to the fear of crime (*Rolf & Stal, 1994*). Therefore, tree density is one of the factors influencing the sense of safety (*Kuo, Bacaicoa & Sullivan, 1998*; *Kuo & Sullivan, 2001*).

## The impact of nature perception on children's development

Since the number of city dwellers increases, it is no surprise that most children play indoors. In several cities in England, there are campaigns like Natural England that aim at doubling the number of farm visits by school children. Actions such as the launch of a new interactive website for families interested in wildlife encourage more children to visit national nature reserves and ensure more people from deprived communities gain access to the natural world. The organisers of the campaigns claim that children are being denied 'the fundamental sense of independence and freedom in nature that their parents enjoyed and the natural environment is there to be explored by children' (*Gray, 2019*).

*Louv (2013)* observes that people love trees but limit children's contact with nature, instilling a kind of ecophobia in them. This results in 'nature deficit disorder' caused by alienation from the natural world. It is crucial not to shield children from outdoor adventures, which, as Louv believes, can be a preventive, additional, and in some cases an alternative cure for many diseases. A close relationship with nature can be helpful in treating depression, mental problems and ADHD syndrome. Contact with nature eliminates stress and increases self-esteem. In open spaces it is easier to establish and deepen social ties. People who need time of 'healthy' loneliness, in the bosom of nature can calm down, relax, find an internal balance, nature gives solace, a feeling of freedom and privacy, outdoor activity positively influences physical health, helps fight obesity, respiratory diseases, sight and the backbone. Man strives for a connexion with nature through the desire for health, well-being, and greater efficiency (*Nilsson et al., 2018*; *Maas et al., 2009*; *Donovan et al., 2011*; *Turner-Skoff & Cavender, 2019*). The innate desire to come in contact with nature is called biophilia, a genetically coded basic need for life

(*Modrzewski & Szkołut, 2015*). *Kellert (2005)* demonstrate that nature is important to children's development in every major way—intellectually, emotionally, socially, spiritually and physically. Play in nature, particularly during middle childhood, is especially important for developing the capacities for creativity, problem-solving, and emotional and intellectual development (*Kellert, 2005*; *Peper et al., 2007*; *WHO's Regional Office for Europe, 1999*).

There is some ambiguity in respondents' approach to the green public spaces. On the one hand, people are willing to pay more for flats with a view (in Poland flats with a park view are more expensive than those without it by ca. 6–10%) (*Rostkowska, 2018*), but on the other hand, parents discourage children from playing in the open air for fear of getting dirty, having ticks, risk from traffic or abduction and abuse. We are looking at an on-going national movement away from nature-based recreation (*Nader, 2008*), for recreation in controlled conditions on safety-tested prefabricated playground equipment (*Pergams & Zaradic, 2008*). However, free outdoor play (*Ginsburg, 2007*; *Lester & Maudsley, 2006*) is a prerequisite for proper development of the sense of independence, decision-making abilities, resilience, and conflict resolution skills. It is as if we forgot about our own parents' arguments: go and get some vitamin D from the sun—as many children are vitamin D deficient (*Huh & Gordon, 2008*); move and burn some calories—as we observe a growing number of overweight children and adolescents (*Huh & Gordon, 2008*; *Ogden, Caroll & Flegal, 2008*; *Ogden et al., 2006*; *Troiano et al., 1995*); trees are good for you—as street trees may help prevent early childhood asthma (*Lovasi et al., 2008*); get away from TV (*Vandewater et al., 2007*) so that you do not spoil your eyes (*Rose et al., 2008*)—considering that young children are growing up in a media-saturated environment; go outside and learn more about nature (*Bebbington, 2005*).

## Differences between urban and rural parents towards greenery

In the metropolitan areas, the residents in general appreciated trees, but age, education, income and perception of vegetation in general affect attitudes towards trees.
The awareness that trees raise the value of property rises with age, education level and affluence of the respondents (*Shirazi & Kazmi, 2016*). Citizens prefer trees near houses and liked both native and exotic tree species, but not those that dropped leaves or tended to fall. The studies show that people simultaneously agree that trees in urban areas cause damage (accidents, infrastructure damage) and offer a lot of benefits (oxygen supply, shade). Overall, respondents agree that there should be more trees in the cities (*Camacho-Cervantes et al., 2014*). Citizens also reveal a consumptive approach to greenery. They would like to benefit from nice views of nature and at the same time they do not want to take direct responsibility for the nature, start caring for the trees or bare costs for their maintenance (*Wojnowska-Heciak, 2019*).

Suburbs being 'half-world between city and country' (*Boyd, 1952*) are expected to offer more green spaces and more trees, which may ensure a different perspective on trees. However, the study with suburban backyarders revealed varied opinions on trees in their backyards. For example the balance between removal and aesthetics includes having to remove beautiful trees that died, as well as choosing to remove ugly ones. Similarly, the

trade-off between tree removal and shade includes situations where people exhibited dubious attitudes. On the one hand, people opted for removing dangerous trees but felt bad about the loss of shade, on the other hand they wanted the trees removed for creating too much shade (*Head et al., 2005*). It seems that occupants of suburbs more often have an even more ambivalent attitude towards trees than city residents. Another study, concerning perception of trees by different suburb types within multiple towns, also showed that most respondents had positive perceptions about urban trees and attached a great deal of importance to those elements of the urban ecosystem (*Gwedla & Shackleton, 2019*).

There is evidence that farmers also have diverse attitudes towards trees growing at their private plots. Views which reflect a lack of trust in the tree management agencies are particularly strong. This can be linked to a need for security, for people to feel safe in their backyards (*Zubair & Garforth, 2006*).

## Childhood in the urban and rural area

The findings of comparative studies describing the childhood in the city and in the countryside, indicating that growing up in the countryside is healthier for children and more stimulating for their physical outdoor activity than living in the city, are not always so evident. The survey conducted on children (1,687 boys; 1,729 girls) recruited from fourth, fifth and sixth grade classes in schools from urban areas, small cities, and rural areas shows obesity was higher among rural children (25%; $P < 0.001$) than in children from urban areas (19%) and small cities (17%). Urban children were the least active (Cohens' $d = -0.4$), particularly around lunchtime while at school ($d = -0.9$ to $-1.1$). Children from small cities exhibited the highest levels of physical activity (*Joens-Matre et al., 2008*).

A study aimed at examining physical activity in rural and urban neighbourhoods shows that urban adults took more steps a day and reported more walking and cycling for transport in the neighbourhood, more recreational walking in the neighbourhood, and more walking for transportation outside the neighbourhood than rural adults. Rural adults reported more recreational cycling in the neighbourhoods (*Dyck et al., 2011*). Therefore, city life does not always mean sedentary life in comparison to rural context. There is evidence that rural residents experience a health disadvantage compared to urban residents, associated with a greater prevalence of health risk factors and socioeconomic differences. Research in Canada identifies higher mortality rates, decreased life expectancy, greater incidence and prevalence of morbidity, and poorer self-reported health status in rural populations. For example life expectancy at birth is at least 2 years less for men in rural areas compared to urban areas and the risks of death from circulatory disease or respiratory disease are as much as 10% higher in rural areas. This health disparity may be the result of different health risk factors, including health behaviours and socioeconomic status (SES). Additionally, different activity patterns between urban and rural populations may potentially lead to differences in exposure and risk(s) related to environmental contaminants, further contributing to the health disparity (*Matz, Stieb & Brion, 2015*). Daily activity patterns analysed in 2010 on young people shows that they get less than 60 min of daily moderate-to-vigorous physical activity (MVPA). Studies suggest that rural adults are less active than their urban counterparts, although studies of children

are equivocal. The relationship between urban and rural residence and MVPA of young people appears to be more complex than previously envisioned. Rural residence appears to be supportive of MVPA in girls but not boys (*Moore et al., 2014*).

In the survey conducted in Cyprus, parents of children in rural schools report more space available in the garden and in the neighbourhoods, and safer neighbourhoods than parents of children in urban schools, whereas children in urban schools have more exercise equipment available at home and are transported more frequently to places where they could be physically active (*Loucaides, 2004*). There are also certain problems of growing up in the countryside. The survey conducted in rural Northamptonshire explores the ways in which children encounter the countryside through their own experiences, and (re)examine the 'rural' from their own viewpoint. The findings uncovered an alternative geography of exclusion and disenfranchisement. Rather than being part of an ideal community many children, especially the least affluent and teenagers, felt dislocated and detached from village life (*Matthews et al., 2000*).

City children do not have ideal rates of physical activity. The study of 40 6- to 8-year-old Hong Kong children reveals that limited availability of outdoor play areas during the afternoon hours resulted in children spending 72.4% of their time sitting and lying down, being active only 10.4% of their time (*Johns & Ha, 1999*). Time and frequency of playing outside were the significant correlates of physical activity in the studies of 4-year-old children (*Baranowski et al., 1993*; *Sallis, 1998*). Results show that 25% of the variance in physical activity was explained by demographic, social and environmental variables, and in the former study, gender, month and location (inside and outside the house) explained 75% of the variance in physical activity. Time spent outdoors and availability of space in the close distance are particularly important for younger children as they need to depend on other people for their transportation to places where they can be physically active. For example *Sallis (1998)* and *Sallis et al. (2001)* finds that parents transporting their children to exercise facilities were a variable significantly associated with physical activity.

## Differences between contemporary and previous generations

There are many ways in which this generation's childhood is different from that of the last generation, but one of the most abrupt contrasts is the degree to which it is being spent indoors. There are lots of reasons, including the marked increase in time spend interacting with electronic devices, the emphasis on scheduled activities and achievements, concerns about sun exposure—and, for many families, the lack of safe outdoor places to play. It is not just children; adults are spending less time outdoors as well (*McCarthy, 2019*). However, children were found to spend less time playing outdoors than their mothers did when they were young—even in rural areas. According to a survey covering over 800 mothers, children in the early 2000s play less outdoors as compared to a generation ago when the mothers interviewed were children: 71% of today's mothers said they recalled playing outdoors every day as children, but only 26% of them said their kids play outdoors daily. A comparison between the children in the early 2000s and those a generation ago indicates clearly that they: (1) spend less time playing outdoors; (2) participate in different activities outdoors (e.g. fewer street games and more organised

youth sports); and (3) participate in more indoor than outdoor play activities. When mothers were asked about obstacles to outdoor play and their thoughts regarding the benefits of outdoor play, almost all of them recognised some of the diverse benefits of outdoor play but also obstacles: such as television, computers, and concerns about crime, safety and injury, prevented their children from participating in more outdoor play. The responses did not vary a great deal between mothers living in rural and urban areas. The results of that study negate the assumption that children living in rural areas would have access to greater public space for play and recreation. They find that farmlands, with their restricted use and lack of local supervision for children's activities, do not offer the rural child more opportunities for outdoor experiences (*Clements, 2004*).

### Aim of the study

The main aim of our research was to compare the attitudes towards trees of two groups of parents: the ones living in a city and those in the suburbs in a rural community in Poland. On the basis of the literature review and our own study, we aimed at understanding how urban and rural residents perceive tree benefits and tree threats, whether the place of residence impacts this perception, and how the parents' general perception of trees translates into their approach to children' play among trees.

## MATERIALS AND METHODS

### Respondents

The study was conducted with a sample survey of parents from two schools in the Mazowieckie Voivodship: Tadeusz Kościuszko Public Primary School in Jedlińsk (52 respondents) and Maria Kann Primary School No. 319 in Warsaw—Ursynów District (59 respondents). These are district public schools in which most of the children come from the closest vicinity. This feature allowed us to collect data from residents from one of the districts of the capital city of Poland (Warsaw) and residents of the suburbs of the medium-size city (Radom) in Poland, which in terms of administrative division is a rural community (Jedlińsk).

In 2018, there were 811 pupils in the primary school in Warsaw, including 363 pupils in 4–6 grade classes, and 450 pupils in the school in Jedlińsk, including 228 in 4–6 grades. The representative for the school sample of parents have children in 4–6 grades classes of primary schools (children ageing from 10 to 12 years). The survey was conducted at the end of the teacher-parent meeting. Most of the respondents were women (80%), probably because mainly women attend school meetings. With regard to the age criteria, people in the range of 30–45 years (72.1%) dominated, 18.9% were under 30, 7.2% in the age group 46–60 and 1.8% over 60 years. Most of the respondents had a university degree (60.4%), 35.1% completed secondary education and 4.5% only had elementary education. The ratio of people living in the countryside and in the city is 52 (46.8%) to 59 (53.2%). To be precise, three people living in the village come from a city, and five people living in the city come from a village. The survey included questions about the place of residence. Detailed sociodemographic characteristics of the respondents are presented in Table 1.

**Table 1 Sociodemographic characteristics of 111 parents participating in the survey.**

|  |  | N | % |  |  | n | % |
|---|---|---|---|---|---|---|---|
| Sex | Male | 22 | 20 | Education | Elementary | 5 | 5 |
|  | Female | 89 | 80 |  | Secondary | 39 | 35 |
| Age | Below 30 | 21 | 19 |  | Higher | 67 | 60 |
|  | 30–45 | 78 | 70 | Place of living | Village | 52 | 67 |
|  | 46 – 60 | 10 | 9 |  | City | 59 | 53 |
|  | Over 60 | 2 | 2 | Place of birth | Village | 54 | 49 |
|  |  |  |  |  | City | 57 | 51 |

The study was performed with respect to the ethical principles of market research and public opinion as defined in the International Code of Market and Social Research, which was developed jointly by the European Association of Public Opinion and Market Researchers, the European Society for Opinion and Marketing Research (ESOMAR), and the International Chamber of Commerce (ICC). study also adheres to the Researchers Quality Control Programme (PKJPA), which was developed and implemented in the Polish market by 'Rynek' public opinion research organisation and adopted by Polish polling organisations.

The survey was performed as a voluntary anonymous questionnaire form given at the end of the parent-teacher meetings to parents. Respondents completed the questionnaire at school. The aim of the research was presented to the participants. Returning completed forms was understood as a consent from the respondents to take part in the research.

## Questionnaire

The questionnaire used in the study was divided into four parts. The first group of questions (basic questions) was built on the principle of 37 sentences about mature trees. The responses were presented on a seven-point Likert scale (anchored by disagreement 'definitely not' and agreement 'absolutely yes') to assess the attitude to a given statement. The statements on benefits and disservices of trees were based on criteria presented by *Roy, Byrne & Pickering (2012)*. They were divided into five categories: social, economic, environmental, health and aesthetic/visual (Table 2).

The second part consisted of seven questions referring to preferences concerning trees and nature. The respondents were asked to assess whether the presence of trees was a significant criterion when choosing a place to live. They also assessed the quality of greenery around the place of residence, as well as the number of trees around it. One of the questions was on opinions regarding contact of children with trees. The respondents also stated their aesthetic preferences concerning plants in public spaces (deciduous trees, coniferous trees, shrubs, perennials and decorative grasses) (Table 3).

In the third group of statements the respondents revealed their attitudes towards tree management. Five statements were included in this part of the questionnaire. Respondents expressed their opinion on the law regarding tree logging, social responsibility for trees, and their own activities in designing green areas (e.g. through public consultations)

**Table 2  Benefits and disservices (problems, hazard costs and expenditures) of trees divided into 5 categories (based on criteria presented by Roy, Byrne & Pickering (2012).**

| | Social | Economic | Environmental | Health | Aesthetic/ Visual |
|---|---|---|---|---|---|
| Benefits | | | | | |
| Making urban environment more pleasant to live, work and spend leisure time | + | | | | |
| Increasing social cohesion | + | | | | |
| Reducing crime | + | | | | |
| Providing nature in the city | + | | | | |
| Reducing speed (drivers are more careful) | + | | | | |
| Savings on electricity costs | | + | | | |
| Increasing property/land value | | + | | | |
| Contributing to the economic vitality of the city | | + | | | |
| Reducing expenditure on road surface repairs | | + | | | |
| Modifying microclimate | | | + | | |
| Improving air quality | | | + | | |
| Reducing rate of storm water runoff | | | + | | |
| Controlling wind | | | + | | |
| Reducing solar radiation in summer | | | + | | |
| Providing shade on sunny days | | | + | | |
| Providing habitat for wildlife | | | + | | |
| Reducing noise | | | + | | |
| Creating relaxed psychological states | | | | + | |
| Reducing stress | | | | + | |
| Looking beautiful while blooming | | | | | + |
| Improving scenic quality | | | | | + |
| Providing privacy | | | | | + |
| Creating seasonal interest by highlighting seasonal changes | | | | | + |
| Cover unpleasant views | | | | | + |
| Disservices (problems, hazard costs and expenditures) | | | | | |
| Causing fear of crime | + | | | | |
| Contributing to road accidents | + | | | | |
| Causing danger on playgrounds | + | | | | |
| Costs of maintaining mature trees overwhelm their benefits | | + | | | |
| Destroying road surface by roots | | + | | | |
| Dropping flowers | | | + | | |
| Dropping leaves | | | + | | |
| Dropping branches and seeds | | | + | | |
| Increasing attack by associated insects | | | | + | |
| Drip sap or sticky residue on parked Cars | | | | | + |
| Obscuring good views | | | | | + |
| Causing darkness | | | | | + |
| Look ugly if not maintained | | | | | + |

**Table 3 Green space in square meters per inhabitant in Warsaw (*Green Areas Per Inhabitant in Warsaw, Poland, 2018*).**

| Land use type in Warsaw | Square metres per inhabitant in Warsaw |
| --- | --- |
| Forest and Woodland | 41.98 |
| Farmland | 22.07 |
| Nature Reserve | 19.76 |
| Maintained Grass | 15.84 |
| Park | 8.32 |
| Public Garden | 0.61 |
| Orchard | 0.36 |
| Golf Course | 0.04 |

**Table 4 Green space in square meters per inhabitant in *Gmina Jedlińsk (2019)*.**

| Land use type in Jedlińsk | Square metres per inhabitant in Jedlińsk |
| --- | --- |
| Farmland | 74.96 |
| Meadows | 11.7 |
| Forests | 11.34 |
| Pasture | 6.9 |
| Orchards | 2.64 |

(Table 4). In the second and third part of the survey the answers were marked on a 5- and 7-point Likert scale (statements regarding trees issues had a 7-point scale, for greater accuracy, and the 5-point scale was adopted in questions regarding respondents' profile). In the last part of the questionnaire, the respondents provided information on their gender, age, education, place of birth and residence (village or city). Respondents were given the opportunity to state their opinion on trees and nature in general. Questionnaires were distributed to parents/guardians during parent-teacher meetings. The test took about 10 min and was anonymous.

## Data analysis

Several methods were used to analyse the results of the study: independence test $\chi^2$, analysis of the distribution of individual responses and their percentage ratio. The validity of choosing the independence test $\chi^2$ was confirmed by the diagram presented in the article of *Wątroba (2019)*. We compared two independent groups (village/city). We tested differences between nominal variables. The purpose of the test was to determine the relationship between two characteristics. Two hypotheses, H0: variables $X$ and $Y$ are independent and H1: variables $X$ and $Y$ are not independent, were formulated. The adopted level of significance (probability level) was $\alpha = 0.05$. If $p < \alpha$, H0 was rejected in favour of the alternative hypothesis. If $p \geq \alpha$, there were no grounds to reject the H0 hypothesis. The $p$-value and the value of the $\chi^2$ test show the degree of dependence between two variables (*Górniak & Wachnicki, 2004*).

The test was performed using the RStudio Desktop and the R Commander (Rcmdr) package (Version 2.6.0) (*Fox, 2005*). The report generated from RStudio allowed determining the relationships between the responses obtained from urban and rural residents.

## Study areas

Warsaw is located on the Vistula River in east-central Poland. Its population is officially estimated at 1.78 million residents (on 51,724 sq. km) within a greater metropolitan area of 3.1 million residents, which makes Warsaw the biggest city in Poland and the 8th most populated capital city in the European Union (*Wikipedia, 2019*; *World Cities, 2019*). According to the data published by TravelBird for 2018, the overall green space per inhabitant in the city of Warsaw was 109.45 square metres, with 41.98 square metres of forest and woodland per person (Table 3) (*Green Areas Per Inhabitant in Warsaw, Poland, 2018*).

Jedlińsk is a rural community functioning as suburbs of Radom, Poland. It is a historic settlement located 90 km south of Warsaw and 10 km north of Radom, on the Radomka and the Tymianka rivers. It had enjoyed town privileges for over 250 years until 1,869 when it was transformed into a farming settlement. Today Jedlińsk has 1.700 inhabitants. The total area of the commune is 138.72 sq. km, and is inhabited by approx. 14,400 people. The commune consists of 31 villages (*Gmina Jedlińsk, 2019*). The overall green space per person in Jedlińsk was 107.54 square metres (Table 4) in 2018 with 74.96 square metres of farmland per inhabitant. The comparison of the green space per inhabitant between Warsaw and Jedlińsk (Tables 3 and 4) is surprising, as the area of forest per inhabitant in Warsaw is almost four times the corresponding area in Jedlińsk. This indicates potentially greater contact with clusters of trees in the city than in the rural commune. The overall green space per resident is also higher in Warsaw than in Jedlińsk, although the difference is not big.

# RESULTS

## General perception of trees: benefits and disservices

In the category of social statements (Table 5), the difference appeared among the benefits. 'Making urban environment more pleasant to live, work and spend leisure time' was highly assessed by residents of the city: almost 75% of them chose the answer 'absolutely yes'. None of the city dwellers chose the answers 'definitely not' and 'no'. Most people did not perceive trees as factors contributing to crime reduction or fear of crime.

Almost all respondents agree that mature trees bring the world of nature closer to people. The most frequently indicated answers were: 'absolutely yes' (city: 59.3%, village: 36.5%) and 'yes' (city: 33.9%, village: 48.1%). In the group of economic statements statistically significant differences appeared among both benefits and disservices (Table 6). Unlike city dwellers, villagers think that trees reduce expenditure on road surface repairs. On the other hand, compared to rural residents, most parents living in the city do not agree with the statement that costs of maintaining mature trees overwhelm their benefits. City inhabitants most often indicated the answer 'not at all' (33.9%) and 'no' (27.1%), while

**Table 5 Differences in perception of social benefits and disservices of trees between rural and urban residents.**

| | | Definitely not (%) | No (%) | Rather no (%) | Do not know (%) | Rather yes (%) | Yes (%) | Absolutely yes (%) | χ² | p-Value |
|---|---|---|---|---|---|---|---|---|---|---|
| Benefits | | | | | | | | | | |
| Making urbanised environment* more pleasant to live, work and spend leisure time | Village | 1.9 | 1.9 | 0 | 0 | 13.5 | 36.5 | 46.2 | 19.098 | 0.004 |
| | City | 0 | 0 | 1.7 | 3.4 | 0 | 20.3 | 74.6 | | |
| Building stronger sense of community | Village | 0 | 5.8 | 9.6 | 50.0 | 9.6 | 13.5 | 11.5 | 4.353 | 0.629 |
| | City | 5.1 | 5.1 | 11.9 | 42.4 | 10.2 | 8.5 | 16.9 | | |
| Reducing crime | Village | 17.3 | 7.7 | 13.5 | 57.7 | 1.9 | 1.9 | 0 | 4.729 | 0.579 |
| | City | 23.7 | 10.2 | 5.1 | 54.2 | 1.7 | 1.7 | 3.4 | | |
| Providing nature in the city | Village | 0 | 0 | 0 | 5.8 | 9.6 | 48.1 | 36.5 | 3.437 | 0.179 |
| | City | 0 | 0 | 0 | 0 | 6.8 | 33.9 | 59.3 | | |
| Reducing speed (drivers are more careful) | Village | 1.9 | 5.8 | 7.7 | 25.0 | 32.7 | 23.1 | 3.8 | 9.228 | 0.161 |
| | City | 8.5 | 5.1 | 18.6 | 23.7 | 20.3 | 13.6 | 10.2 | | |
| Disservices (problems, hazard costs and expenditures) | | | | | | | | | | |
| Causing fear of crime | Village | 13.5 | 7.7 | 23.1 | 46.2 | 3.8 | 3.8 | 1.9 | 4.757 | 0.575 |
| | City | 11.9 | 13.6 | 23.7 | 32.2 | 11.9 | 3.4 | 3.4 | | |
| Contributing to road accidents | Village | 17.3 | 15.4 | 34.6 | 5.8 | 17.3 | 9.6 | 0 | 5.380 | 0.496 |
| | City | 23.7 | 16.9 | 33.9 | 8.5 | 10.2 | 3.4 | 3.4 | | |

**Note:**

* Urbanized environment meaning public spaces or common spaces that have features of build-up areas that can be found in the city, in the suburbs or in the countryside.

**Table 6 Differences in perception of economic benefits and disservices between rural and urban residents.**

| | | Definitely not (%) | No (%) | Rather no (%) | Do not know (%) | Rather yes (%) | Yes (%) | Absolutely yes (%) | χ² | p-Value |
|---|---|---|---|---|---|---|---|---|---|---|
| Benefits | | | | | | | | | | |
| Increasing property/land value | Village | 3.8 | 0 | 11.5 | 32.7 | 19.2 | 17.3 | 15.4 | 5.668 | 0.461 |
| | City | 0 | 1.7 | 10.2 | 23.7 | 23.7 | 15.3 | 25.4 | | |
| Savings on electricity costs | Village | 1.9 | 5.8 | 11.5 | 48.1 | 7.7 | 7.7 | 17.3 | 3.415 | 0.755 |
| | City | 1.7 | 8.5 | 5.1 | 44.1 | 15.3 | 10.2 | 15.3 | | |
| Contributing to the economic vitality of the city | Village | 3.8 | 0 | 5.8 | 30.8 | 15.4 | 30.8 | 13.5 | 3.8983 | 0.1424 |
| | City | 1.7 | 0 | 0 | 27.1 | 15.3 | 27.1 | 28.8 | | |
| Reducing expenditure on road surface repairs | Village | 0 | 3.8 | 15.4 | 53.8 | 9.6 | 11.5 | 5.8 | 15.545 | 0.016 |
| | City | 6.8 | 15.3 | 8.5 | 59.3 | 0 | 5.1 | 5.1 | | |
| Disservices (problems, hazard costs and expenditures) | | | | | | | | | | |
| Costs of maintaining of mature trees overwhelm their benefits | Village | 17.3 | 15.4 | 26.9 | 30.8 | 3.8 | 3.8 | 1.9 | 13.805 | 0.032 |
| | City | 33.9 | 27.1 | 10.2 | 13.6 | 5.1 | 8.5 | 1.7 | | |
| Destroying road surface by roots | Village | 3.8 | 13.5 | 30.8 | 26.9 | 17.3 | 3.8 | 3.8 | 10.831 | 0.094 |
| | City | 16.9 | 16.9 | 23.7 | 10.2 | 25.4 | 5.1 | 1.7 | | |

**Table 7 Differences in perception of environmental benefits and disservices between rural and urban residents.**

| | | Definitely not | No | Rather no | Do not know | Rather yes | Yes | Absolutely yes | $\chi^2$ | p-Value |
|---|---|---|---|---|---|---|---|---|---|---|
| Benefits | | | | | | | | | | |
| Modifying microclimate | Village | 0 | 1.9 | 1.9 | 1.9 | 7.7 | 42.3 | 44.2 | 2.240 | 0.326 |
| | City | 0 | 0 | 0 | 3.4 | 5.1 | 15.3 | 76.3 | | |
| Improving air quality | Village | 0 | 0 | 0 | 3.8 | 5.8 | 30.8 | 59.6 | 1.134 | 0.509 |
| | City | 0 | 1.7 | 0 | 1.7 | 10.2 | 23.7 | 62.7 | | |
| Reducing rate of storm water runoff | Village | 1.9 | 3.8 | 7.7 | 40.4 | 11.5 | 15.4 | 19.2 | 3.288 | 0.772 |
| | City | 3.4 | 3.4 | 6.8 | 40.7 | 5.1 | 25.4 | 15.3 | | |
| Controlling wind | Village | 3.8 | 1.9 | 3.8 | 7.7 | 25.0 | 38.5 | 19.2 | 6.333 | 0.387 |
| | City | 0 | 1.7 | 1.7 | 6.8 | 42.4 | 27.1 | 20.3 | | |
| Reducing solar radiation in summer | Village | 0 | 1.9 | 3.8 | 9.6 | 23.1 | 38.5 | 23.1 | 2.146 | 0.342 |
| | City | 0 | 1.7 | 1.7 | 18.6 | 23.7 | 23.7 | 30.5 | | |
| Providing shade on sunny days | Village | 0 | 1.9 | 1.9 | 5.8 | 1.9 | 25.0 | 63.5 | 2.412 | 0.878 |
| | City | 1.7 | 1.7 | 0 | 6.8 | 1.7 | 20.3 | 67.8 | | |
| Providing habitat for wildlife | Village | 0 | 0 | 5.8 | 23.1 | 13.5 | 28.8 | 28.8 | 1.135 | 0.567 |
| | City | 0 | 1.7 | 1.7 | 16.9 | 11.9 | 30.5 | 37.3 | | |
| Reducing noise | Village | 0 | 0 | 1.9 | 7.7 | 15.4 | 44.2 | 30.8 | 1.124 | 0.570 |
| | City | 0 | 0 | 1.7 | 13.6 | 20.3 | 28.8 | 35.6 | | |
| Disservices (problems, hazard costs and expenditures) | | | | | | | | | | |
| Dropping flowers | Village | 11.5 | 34.6 | 28.8 | 3.8 | 11.5 | 7.7 | 1.9 | 6.038 | 0.419 |
| | City | 27.1 | 27.1 | 23.7 | 6.8 | 8.5 | 3.4 | 3.4 | | |
| Dropping leaves | Village | 3.8 | 13.5 | 42.3 | 3.8 | 11.5 | 25.0 | 0 | 20.142 | 0.003 |
| | City | 23.7 | 16.9 | 27.1 | 0 | 18.6 | 8.5 | 5.1 | | |
| Dropping branches and seeds | Village | 5.8 | 26.9 | 30.8 | 5.8 | 21.2 | 7.7 | 1.9 | 11.868 | 0.065 |
| | City | 27.1 | 18.6 | 23.7 | 1.7 | 13.6 | 10.2 | 5.1 | | |

rural residents chose 'rather not' (26.9%) and 'I do not know' (30.8%). It turns out that city dwellers are less inclined to say that the cost of maintaining the trees exceeds their benefits.

Environmental statements constituted the biggest category of statements. But again, only one of them showed a statistically significant difference between rural and urban parents (Table 7). Around 40% of city dwellers did not perceive 'dropping leaves' as annoyance, while 25% of villagers chose the answer 'yes' when evaluating the statement.

Most respondents agree that mature trees create a favourable microclimate. The most frequently chosen answer was 'absolutely yes' (city: 76.3%, village: 44.2%) and yes (city: 15.3%, village: 42.3%). Two people living in the countryside did not agree with it, and one had no opinion. The respondents agree with the statement that mature trees significantly improve air quality. Only three people had no opinion and one person did not agree. The most frequently chosen option was 'Absolutely yes' (city: 62.7%, village: 59.6%). There are no significant differences in response behaviour and resident group. The majority

**Table 8 Differences in perception of health benefits and disservices between rural and urban residents.**

| | | Definitely not | No | Rather no | Do not know | Rather yes | Yes | Absolutely yes | $\chi^2$ | p-Value |
|---|---|---|---|---|---|---|---|---|---|---|
| **Benefits** | | | | | | | | | | |
| Creating relaxed psychological states | Village | 0 | 1.9 | 1.9 | 1.9 | 9.6 | 28.8 | 55.8 | 1.609 | 0.447 |
| | City | 0 | 1.7 | 0 | 0 | 10.2 | 11.9 | 76.3 | | |
| Reducing stress | Village | 0 | 0 | 1.9 | 3.8 | 11.5 | 32.7 | 50.0 | 1.054 | 0.590 |
| | City | 0 | 1.7 | 0 | 8.5 | 3.4 | 18.6 | 67.8 | | |
| **Disservices (problems. hazard costs and expenditures)** | | | | | | | | | | |
| Increasing attack by associated insects | Village | 0 | 9.6 | 25.0 | 25.0 | 26.9 | 11.5 | 1.9 | 11.294 | 0.080 |
| | City | 15.3 | 18.5 | 13.6 | 30.5 | 18.6 | 10.2 | 3.4 | | |

of respondents do not feel the threat from trees with fragile branches. The most frequently chosen answer categories were 'rather not' (village: 21.2%, city: 23.7%), 'I do not know' (village: 11.5%, city: 15.3%) and 'rather yes' (village: 28.8%, city: 13.6%)

No differences between resident groups were found for the items in the category of health benefits and disservices (Table 8). Most respondents agree with the statement that the presence of trees has a soothing effect on nerves and stress (67.8% of urban residents and 50.0% of rural residents). Both groups think that the presence of trees helps to create relaxed psychological states (55.8% of villagers and 76.3 of city dwellers). In response to 'increasing attack by associated insects,' most respondents chose statements ranging from 'rather no' to 'rather yes.'

Only one statement from the category of aesthetic disservices of trees was differently evaluated by urban and rural parents (Table 9). Almost 45% of city residents did not agree with the statement that trees 'look ugly if not maintained.' Less than 20% of villagers shared this opinion. Most of the respondents strongly appreciated benefits of trees such as: 'looking beautiful while blooming' and 'creating seasonal interest by highlighting seasonal changes.'

## Preferences towards outdoor activity of children

The way parents perceive trees affects the outdoor activity of children (*Tappe et al., 2013*). And, at the same time, it is the surroundings that contribute to shaping us and our perception of the world (*Ellard, 2015*). All this seems particularly important when taking into account the undergoing degradation of the natural environment, decreasing number of trees in cities, and, most of all, the need to increase the environmental awareness among young generations. Therefore, attention the residents of Warsaw and Jedlińsk was brought to children's outdoor activities among the trees.

For most of the statements, the answers of city dwellers and villagers were similar (Table 10). In general, respondents believe that mature trees have a positive impact on the development of children. A small percentage of people have no opinion on the subject (village: 5.8%, city: 6.8%). Only one city resident disagrees with this opinion. Most of the respondents believe that mature trees cause allergies. Some people have no opinion on

**Table 9 Differences in perception of aesthetic/visual benefits and disservices between rural and urban residents.**

| | | Definitely not (%) | No (%) | Rather no (%) | Do not know (%) | Rather yes (%) | Yes (%) | Absolutely yes (%) | $\chi^2$ | p-Value |
|---|---|---|---|---|---|---|---|---|---|---|
| Benefits | | | | | | | | | | |
| Looking beautiful while blooming | Village | 0 | 0 | 0 | 1.9 | 5.8 | 23.1 | 69.2 | 1.054 | 0.590 |
| | City | 0 | 0 | 0 | 1.7 | 5.1 | 20.3 | 72.9 | | |
| Improving scenic quality | Village | 0 | 0 | 0 | 1.9 | 13.5 | 46.2 | 38.5 | 1.127 | 0.634 |
| | City | 0 | 0 | 1.7 | 1.7 | 8.5 | 35.6 | 52.5 | | |
| Providing privacy | Village | 1.9 | 0 | 15.4 | 11.5 | 21.2 | 28.8 | 30.2 | 12.508 | 0.051 |
| | City | 1.7 | 1.7 | 5.1 | 0 | 20.3 | 35.6 | 35.6 | | |
| Creating seasonal interest by highlighting seasonal changes | Village | 0 | 0 | 1.9 | 0 | 3.8 | 28.8 | 65.4 | 2.894 | 0.235 |
| | City | 0 | 0 | 0 | 3.4 | 1.7 | 10.2 | 84.7 | | |
| Cover unpleasant views | Village | 0 | 0 | 1.9 | 9.6 | 23.1 | 50.0 | 15.4 | 12.261 | 0.056 |
| | City | 1.7 | 1.7 | 1.7 | 10.2 | 16.9 | 27.1 | 40.7 | | |
| Drip sap or sticky residue on parked cars | Village | 19.2 | 17.3 | 32.7 | 17.3 | 7.7 | 3.8 | 1.9 | 11.341 | 0.078 |
| | City | 15.3 | 28.8 | 15.3 | 8.5 | 20.3 | 8.5 | 3.4 | | |
| Obscuring good views | Village | 7.7 | 21.2 | 26.9 | 3.8 | 21.2 | 11.5 | 7.7 | 9.098 | 0.168 |
| | City | 18.6 | 25.4 | 23.7 | 8.5 | 8.5 | 13.6 | 1.7 | | |
| Causing darkness | Village | 7.7 | 13.5 | 32.7 | 9.6 | 28.8 | 5.8 | 1.9 | 6.336 | 0.387 |
| | City | 8.5 | 27.1 | 30.5 | 10.2 | 15.3 | 8.5 | 0 | | |
| Look ugly if not maintained | Village | 1.9 | 17.3 | 19.2 | 3.8 | 36.5 | 19.2 | 1.9 | 15.101 | 0.019 |
| | City | 22.0 | 22.0 | 13.6 | 6.8 | 20.3 | 10.2 | 5.1 | | |

**Table 10 Differences between urban and rural parents with regard to approaches to children's play in natural surroundings.**

| | | Definitely not | No | Rather no | Do not know | Rather yes | Yes | Absolutely yes | $\chi^2$ | p-Value |
|---|---|---|---|---|---|---|---|---|---|---|
| Improving physical health | Village | 0 | 1.9 | 0 | 13.5 | 15.4 | 32.7 | 36.5 | 6.866 | 0.333 |
| | City | 5.1 | 3.4 | 5.1 | 6.8 | 13.6 | 30.5 | 35.6 | | |
| Improving psychological health | Village | 0 | 0 | 0 | 3.8 | 13.5 | 28.8 | 53.8 | 4.109 | 0.128 |
| | City | 0 | 1.7 | 5.1 | 1.7 | 11.9 | 22.0 | 57.6 | | |
| Improving children's development | Village | 0 | 0 | 0 | 5.8 | 21.2 | 34.6 | 38.5 | 0.9479 | 0.6225 |
| | City | 0 | 0 | 1.7 | 6.8 | 8.5 | 23.7 | 59.3 | | |
| Causing danger on playgrounds (and therefore should be removed) | Village | 15.4 | 30.8 | 21.2 | 5.8 | 13.5 | 7.7 | 5.8 | 17.95 | 0.006 |
| | City | 42.4 | 28.8 | 11.9 | 10.2 | 1.7 | 5.1 | 0 | | |
| Breakable branches are a threat to safety | Village | 3.8 | 23.1 | 21.2 | 11.5 | 28.8 | 11.5 | 0 | 7.157 | 0.307 |
| | City | 10.2 | 23.7 | 28.8 | 15.3 | 13.6 | 6.8 | 1.7 | | |
| Increase allergy attacks | Village | 3.8 | 5.8 | 5.8 | 26.9 | 30.8 | 25.0 | 1.9 | 9.476 | 0.148 |
| | City | 3.4 | 11.9 | 15.3 | 13.6 | 25.4 | 20.3 | 10.2 | | |
| Do you try to protect children from trees (prohibition of climbing trees, touching rotten parts, playing with leaves, branches)? | Village | 3.8 | 23.1 | 23.1 | 7.7 | 9.6 | 21.2 | 11.5 | 12.569 | 0.0504 |
| | City | 23.7 | 20.3 | 18.6 | 3.4 | 8.5 | 8.5 | 16.9 | | |

this subject (village: 26.9%, city: 13.6%) and others do not agree with this statement (15.4% of rural residents and 30.6% of urban residents).

We found a nearly statistically significant difference between rural and urban parents' responses to the question 'Do you try to protect children from trees? (prohibited tree climbing, touching rotten parts, playing with leaves, branches') amounted to 0.0504, which means that the result is on the verge of a significant difference. The biggest differences between urban and rural parents with regard to responses to this question were found for the 'totally no' option (village: 3.8%, city: 23.7%) and 'yes' (village: 21.2%, city: 8.5%).

Rural parents stronger disagreed to the statement '(Trees are) causing danger on playgrounds' compared to urban parents (0.006). This indicates a statistically significant difference between urban and rural parents. The most frequently indicated response of the rural residents was 'no' (30.8%) and 'rather no' (21.2%) and 'definitely not' (42.4%) and 'no' (28.8%) among the urban residents. The results indicate that people living in the countryside feel a greater threat from trees on the playgrounds.

Most respondents agree that mature trees contribute to the improvement of the physical health of people (children). The most frequently chosen response was 'absolutely yes' (city: 35.6%, village: 36.5%) and 'yes' (city: 30.5%, village: 32.7%). Some people did not have an opinion on this subject (city: 6.8%, village: 13.5%). One inhabitant of the village and eight city residents disagreed with this opinion.

Both city residents and villagers ask for more trees, greenery, parks, forests. One respondent commented on the need to protect children from the trees because of sticky resin and ticks.

## DISCUSSION

The survey revealed that urban and rural parents had generally similar views on the presence of trees in the surroundings, the benefits they offer and threats they pose when managed improperly. The research showed that despite the threats that trees pose, they are an invaluable element of surrounding landscape. In three out of five categories of tree benefits and disservices only one statement was evaluated differently by urban and rural parents. Two statistically significant differences appeared in the economic category of statements, and none in the health category (however, the group consisted of three statements only). For most attitudes regarding trees, we did not find differences between urban and rural parents (Tables 5–9). The same observation has been reported in other studies concerning similar issues (Joens-Matre et al., 2008; Dyck et al., 2011; Matz, Stieb & Brion, 2015; Moore et al., 2014; Loucaides, 2004; Matthews et al., 2000; Johns & Ha, 1999; Baranowski et al., 1993; Sallis, 1998; Sallis et al., 2001).

The results obtained in the present survey confirmed that despite the dominating positive perception of trees, parents perceive certain threats or disservices resulting from tree presence (Table 10). Our study showed that villagers are more concerned with 'protecting' children from trees (Table 10, question 'Do you try to protect children from trees? (a ban on tree climbing, touching rotten parts, playing with leaves, branches')). Probably, parents' belief in deleterious impact of trees makes city and suburban children

spend less time playing outside than did children who were born a generation earlier. This corresponds with the findings from other studies (*McCarthy, 2019*; *Clements, 2004*).

The results obtained in Warsaw and Jedlińsk show only few statistically significant differences regarding the attitude towards trees (Table 10). City respondents more often agreed to the statement that mature trees increase the comfort of staying in public places. This may be due to the fact that the villagers living in suburban areas are assumed to have more contact with vegetation and therefore cannot appreciate its presence to such a great extent. Large discrepancies were found regarding the approach to old, damaged trees. Most of the rural residents considered them as an unattractive element of the landscape, whereas people living in cities did not share this opinion. Urban residents have more positive attitudes towards trees, have higher appreciation for the benefits of their presence, feel less threat from trees, and agree that mature trees increase the comfort of visiting public places. They also pay more attention to the presence of trees when choosing a place of residence. Rural residents more often believe that old, damaged trees are visually unattractive. The results of this study clearly suggested that the absence of trees near the residents' homes makes them feel deprived of natural environment and makes them nostalgic for verdant neighbourhoods. It may be that villages have ceased to be as natural as we imagine. It may also be that we all 'come from the city' and Mother Nature seems to intimidate us.

Rural parents feel greater danger posed by trees for their children than the city residents (trees on playgrounds, climbing trees, playing with leaves and branches, etc.) (Table 10). Against the background of the prevalence of attention deficit disorder and the benefits that nature provides, the current concerns about barriers limiting children's access to nature are not unfounded.

However, for most attitudes towards trees, we did not find significant differences between urban and rural parents. Homogeneous results are probably influenced by a similar structure of greenery of the selected settlement units ("Study Areas" Tables 3 and 4) and the proximity of rural Jedlińsk–Radom. On the one hand, Warsaw has never had a compact city structure and included a patchwork of green open spaces. The Białowieża forest and the semi-wild Vistula riverfront constitute the largest forested areas in Warsaw and its surroundings. On the other hand, Jedlińsk is a very closely located suburb of the city of Radom. This may facilitate the migration of people from the city to the immediate vicinity and therefore urban living patterns are maintained. A similar behaviour of rural and urban landscape planners with regard to trees means that the concept expressed in Ildefonso Cerdá's General Theory of 'Urbanización' (making urban rural and rural urban) has already materialised in some places. Perhaps a red lamp should light up for those who deal with spatial planning in terms of further strong urbanisation of suburban areas. It seems that we have reached the point where both environments are similar or homogeneous with respect to the structure of green areas ("Study Areas" Tables 3 and 4) and the way the residents perceive their surrounding space ("Results" Tables 5–9). The sprawling urbanisation process in suburban areas is visible in our social study.

## CONCLUSIONS

The study should be extended to cover the male approach to raising children in the environment that includes trees, especially mature trees. Future research may select a sample which is representative with regard to gender. Maybe the issues of motherly fear for the safety of their offspring could been more balanced by the fathers' views.

The study results confirm that the urbanisation is sprawling and the differences between urban and rural communities tend to diminish. The question is whether the number of trees and the general area of green public spaces will increase in the development plans for cities, suburbs and rural areas; whether the numerous theories about the benefits of trees in the environment, as reflected in the literature, will find practical application and we will see more trees in the neighbourhoods; what kind of new typologies of land development (urban and non-urban) structures will evolve in coming years; whether and how the new typologies would differ in terms of the amount and the layout of the green areas.

## ACKNOWLEDGEMENTS

We thank Mrs. Izabela Konwerska-Barciak Principal of Maria Kann Primary School No. 319 in Warsaw and Mrs. Elżbieta Religa Principal of Tadeusz Kościuszko Public Primary School in Jedlińsk for support to our research.

### Funding

This work was supported by the Kielce University of Technology, Department of Civil Engineering and Architecture. The funders had no role in study design, data collection and analysis, decision to publish, or preparation of the manuscript.

### Grant Disclosures

The following grant information was disclosed by the authors:
Kielce University of Technology, Department of Civil Engineering and Architecture.

### Competing Interests

The authors declare that they have no competing interests.

### Author Contributions

- Magdalena Wojnowska-Heciak analysed the data, authored or reviewed drafts of the paper, and approved the final draft.
- Magdalena Błaszczyk analysed the data, prepared figures and/or tables, and approved the final draft.
- Marzena Suchocka conceived and designed the experiments, authored or reviewed drafts of the paper, and approved the final draft.
- Joanna Kosno-Jończy conceived and designed the experiments, performed the experiments, analysed the data, prepared figures and/or tables, and approved the final draft.

## Human Ethics

The following information was supplied relating to ethical approvals (i.e. approving body and any reference numbers):

Our research was conducted according to the law in force in Poland, in particular regarding the protection of personal data of respondents, in accordance with the Act of 29 August 1997 on the protection of personal data, in accordance with Regulation (EU) 2016/679 of the European Parliament and of the Council of 27 April 2016 on the protection of individuals with regard to the processing of personal data, the free flow of such data. The study was performed with respect to the ethical principles of market research and public opinion as defined in the International Code of Market and Social Research, which was developed jointly by the European Association of Public Opinion and Market Researchers, the European Society for Opinion and Marketing Research (ESOMAR), and the International Chamber of Commerce (ICC).

The study also adheres to the Researchers Quality Control Programme (PKJPA), which was developed and implemented in the Polish market by "Rynek" public opinion research organisation and adopted by Polish polling organisations.

Please see the Code on the market and social research (that acts in Poland) Article 6—where it is stated that all procedures concerning data protection, approvals concern studies were personal data have been collected and we have performed an anonymous voluntary survey without personal data and therefore those procedures do not apply to our case: https://iccwbo.org/publication/iccesomar-international-code-on-market-and-social-research/.

## Data Availability

The raw measurements are available in the Supplemental Files.

## Supplemental Information

Supplemental information for this article can be found online at http://dx.doi.org/10.7717/peerj.8875#supplemental-information.

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
