# Peer review of "Urban–rural differences in perception of trees described by parents bringing up children in Warsaw and Jedlińsk, Poland"

_PeerJ, doi:10.7717/peerj.8875_

## Round 0.1 · original submission · Major Revisions

Thank you for your submission. Two reviewers have checked your manuscript. My decision is that this manuscript needs major revisions. First, you should update your references (see the recommendations provided by the reviewer #2. Secondly, your manuscript needs to be checked by the native English speaker or language services. In addition, reviewer #2 sees that you should restructure chapters for discussion and results. Read carefully attached file provided by the reviewer #2, and consider all suggestions and comments.

Reviewer 1 ·

Basic reporting

The article is in general written in clear, professional English but please check these two sentences:

Page 4, line 74-75, sentence beginning with "Trees conflict with the road infrastructure is one of the main reasons..." Is there something wrong with this sentence? My (non-native) ear says that it ought to be either "Trees' conflict" or "Trees conflicting". Please check this.

Page 12, line 293, sentence "It is evident that city dwellers are more inclined to say that the cost of maintaining the trees exceeds their benefits." Is this in conflict with the findings? Should it be "city dwellers are less inclined"? Please check this.

The discussion is somewhat unusually long and includes quite a lot of new literature. Some of this literature might have been included already in the introduction, but it is not necessary to make any changes.

Experimental design

This is a well designed study. My only comment is that I wish to see a reason for choosing these respondents in chapter 2.1. Why were parents of primary school children the target group, and why were these two schools picked?

Validity of the findings

The conclusions are otherwise well stated, but the word "much" on line 547 (city respondents much more often agreed) seems exaggerated compared to the findings. After all, the main conclusion is that there was no great difference between the two groups.

Additional comments

Thank you for the chance of reading this clear and interesting study.

·

Basic reporting

In my opinion, the manuscript would benefit from proofreading by a native speaker. Your Introduction is well written and comprehensive. However, in my opinion, you should update your references (some research you cite is from the 80s and more recently published papers are available). For example:
• Trees reduce urban heat stress:
Potchter, O., Cohen, P., Bitan, A., 2006. Climatic behavior of various urban parks during hot and humid summer in the Mediterranean City of Tel Aviv, Israel. Int. J. Climatol. 26, 1695–1711, http://dx.doi.org/10.1002/joc
Bowler, D.E., Buyung-Ali, L., Knight, T.M., Pullin, A.S., 2010. Urban greening to cool towns and cities: a systematic review of the empirical evidence. Landsc. Urban Plan. 97 (3), 147–155, http://dx.doi.org/10.1016/j.landurbplan.2010.05.006
Shashua-Bar, L., Potchter, O., Bitan, A., Yaakov, Y., 2010. Microclimate modelling of street tree species effects within the varied urban morphology in the Mediterranean city of Tel Aviv, Israel. Int. J. Climatol. 30, 44–57, http://dx.doi.org/10.1002/joc
McCarthy, M.P., Best, M.J., Betts, R.A., 2010. Climate change in cities due to global warming and urban effects. Geophys. Res. Lett. 37, 1–5, http://dx.doi.org/10. 1029/2010GL042845
• Trees reduces building energy use:
Akbari, H., Pomerantz, M., Taha, H., 2001. Cool surfaces and shade trees to reduce energy use and improve air quality in urban areas. Sol. Energy 70 (3), 295–310, http://dx.doi.org/10.1016/S0038-092X(00)00089-X
Nowak, D.J., Dwyer, J.F., 2007. Understanding the benefits and costs of urban forest ecosystems. In: Kuser, J.E. (Ed.), Urban and Community Forestry in the Northeast (2nd). Springer Netherlands, Dordrecht, pp. 25–46, http://dx.doi.org/10.1007/978-1-4020-4289-82
Escobedo, F.J., Kroeger, T., Wagner, J.E., 2011. Urban forests and pollution mitigation: analyzing ecosystem services and disservices. Environ. Pollut. 159 (8–9), 2078–2087, http://dx.doi.org/10.1016/j.envpol.2011.01.010
• Trees reduce atmospheric carbon dioxide:
Nowak, D.J., Dwyer, J.F., 2007. Understanding the benefits and costs of urban forest ecosystems. In: Kuser, J.E. (Ed.), Urban and Community Forestry in the Northeast (2nd). Springer Netherlands, Dordrecht, pp. 25–46, http://dx.doi.org/10.1007/978-1-4020-4289-82
Escobedo, F.J., Kroeger, T., Wagner, J.E., 2011. Urban forests and pollution mitigation: analyzing ecosystem services and disservices. Environ. Pollut. 159 (8–9), 2078–2087, http://dx.doi.org/10.1016/j.envpol.2011.01.010
• Trees reduce rainfall runoff, flooding, noise levels, and wind speed:
Tyrväinen, L., Pauleit, S., Seeland, K., de Vries, S., 2005. Benefits and uses of urban forests and trees. In: Konijnendijk, C.C., Nilsson, K., Randrup, T.B., Schipperijn, J. (Eds.), Urban Forests and Trees. Springer, Berlin, pp. 81–114.
Nowak, D.J., Dwyer, J.F., 2007. Understanding the benefits and costs of urban forest ecosystems. In: Kuser, J.E. (Ed.), Urban and Community Forestry in the Northeast (2nd). Springer Netherlands, Dordrecht, pp. 25–46, http://dx.doi.org/10.1007/978-1-4020-4289-82
• Trees remove air pollutants:
Nowak, D.J., Hirabayashi, S., Greenfield, E., 2014. Tree and forest effects on air quality and human health in the United States. Environ. Pollut. 193, 119–129, http://dx.doi.org/10.1016/j.envpol.2014.05.028
Nowak, D.J., Dwyer, J.F., 2007. Understanding the benefits and costs of urban forest ecosystems. In: Kuser, J.E. (Ed.), Urban and Community Forestry in the Northeast (2nd). Springer Netherlands, Dordrecht, pp. 25–46, http://dx.doi.org/10.1007/978-1-4020-4289-82
Escobedo, F.J., Kroeger, T., Wagner, J.E., 2011. Urban forests and pollution mitigation: analyzing ecosystem services and disservices. Environ. Pollut. 159 (8–9), 2078–2087, http://dx.doi.org/10.1016/j.envpol.2011.01.010
• Aesthetic value of trees:
Lohr, V.I., Pearson-Mims, C.H., 2006. Responses to scenes with spreading, rounded, and conical tree forms. Environ. Behav. 38 (5), 667–688, http://dx.doi.org/10.1177/0013916506287355
Schroeder, H.W., Flannigan, J., Coles, R., 2006. Residents’ attitudes toward street trees in the UK and U.S. communities. Arboricult. Urban For. 32 (5), 236–246.
• Relief from stress when being surrounded by nature:
Jiang, B., Chang, C.-Y., Sullivan, W.C., 2014a. A dose of nature: tree cover, stress reduction, and gender differences. Landsc. Urban Plan. 132, 26–36, http://dx.doi.org/10.1016/j.landurbplan.2014.08.005
Jiang, B., Li, D., Larsen, L., Sullivan, W.C., 2014b. A dose–response curve describing the relationship between urban tree cover density and self-reported stress recovery. Environ. Behav., 1–23, http://dx.doi.org/10.1177/0013916514552321
Van den Berg, A.E., Jorgensen, A., Wilson, E.R., 2014. Evaluating restoration in urban green spaces: does setting type make a difference? Landsc. Urban Plan. 127, 173–181, http://dx.doi.org/10.1016/j.landurbplan.2014.04.012
• The presence of nearby nature increases property values:
Wolch, J. R., Byrne, J., & Newell, J. P. (2014). Urban green space, public health, and environmental justice: The challenge of making cities ’just green enough’. Landscape and Urban Planning, 125, 234–244, http://dx.doi/10.1016/j.landurbplan.2014.01.017
• Trees contribute to urban neighbourhoods’ aesthetic quality and enhance human mental and physical health and well-being:
Tyrväinen, L., Pauleit, S., Seeland, K., de Vries, S., 2005. Benefits and uses of urban forests and trees. In: Konijnendijk, C.C., Nilsson, K., Randrup, T.B., Schipperijn, J. (Eds.), Urban Forests and Trees. Springer, Berlin, pp. 81–114.
Nilsson, K., Sangster, M., Konijnendijk, C.C., 2011. Forests trees and human health and well-being: introduction. In: Nilsson, K., Sangster, M., Gallis, C., Hartig, T., de Vries, S., Seeland, K., Schipperijn, J. (Eds.), Forests, Trees and Human Health. Springer, Dordrecht, pp. 1–19.
De Vries, S., Verheij, R.A., Groenewegen, P.P., Spreeuwenberg, P., 2003. Natural environments – healthy environments? An exploratory analysis of the relationship between greenspace and health. Environ. Plan. 35 (10), 1717–1731, http://dx.doi.org/10.1068/a35111
Maas, J., Verheij, R.A., Groenewegen, P.P., de Vries, S., Spreeuwenberg, P., 2006. Green space, urbanity, and health: how strong is the relation? J. Epidemiol. Community Health 60 (7), 587–592, http://dx.doi.org/10.1136/jech.2005.043125
Maas, J., Verheij, R.A., de Vries, S., Spreeuwenberg, P., Schellevis, F.G., Groenewegen, P.P., 2009. Morbidity is related to a green living environment. J. Epidemiol. Community Health 63 (12), 967–973, http://dx.doi.org/10.1136/jech.2008.079038
Mitchell, R., Popham, F., 2007. Greenspace, urbanity and health: Relationships in England. J. Epidemiol. Community Health 61 (8), 681–683, http://dx.doi.org/10.1136/jech.2006.053553
Taylor, M.S., Wheeler, B.W., White, M.P., Economou, T., Osborne, N.J., 2015. Research note: urban street tree density and antidepressant prescription rates – a cross-sectional study in London, UK. Landsc. Urban Plan. 136, 174–179, http://dx.doi.org/10.1016/j.landurbplan.2014.12.005
Donovan, G.H., Michael, Y.L., Butry, D.T., Sullivan, A.D., Chase, J.M., 2011. Urban trees and the risk of poor birth outcomes. Health Place 17 (1), 390–393, http://dx.doi.org/10.1016/j.healthplace.2010.11.004
Hansmann, R., Hug, S.-M., Seeland, K., 2007. Restoration and stress relief through physical activities in forests and parks. Urban For. Urban Green. 6 (4), 213–225, http://dx.doi.org/10.1016/j.ufug.2007.08.004
Korpela, K., Borodulin, K., Neuvonen, M., Paronen, O., Tyrväinen, L., 2014. Analyzing the mediators between nature-based outdoor recreation and emotional well-being. J. Environ. Psychol. 37, 1–7, http://dx.doi.org/10.1016/j.jenvp.2013. 11.003
Berman, M.G., Jonides, J., Kaplan, S., 2008. The cognitive benefits of interacting with nature. Psychol. Sci. 19 (12), 1207–1212, http://dx.doi.org/10.1111/j.1467-9280.2008.02225.x
Tyrväinen, L., Ojala, A., Korpela, K., Lanki, T., Tsunetsugu, Y., Kagawa, T., 2014. The influence of urban green environments on stress relief measures: a field experiment. J. Environ. Psychol. 38, 1–9, http://dx.doi.org/10.1016/j.jenvp.2013.12.005
• Trees and nature are associated with social cohesion:
Holtan, M.T., Dieterlen, S.L., Sullivan, W.C., 2014. Social life under cover: tree canopy and social capital in Baltimore, Maryland. Environ. Behav. 46 (6), 1–24, http://dx.doi.org/10.1177/0013916513518064
• Review articles on positive effects of nearby urban forest resources on human health and well-being:
Keniger, L.E., Gaston, K.J., Irvine, K.N., Fuller, R.A., 2013. What are the benefits of interacting with nature? Int. J. Environ. Res. Public Health 10 (3), 913–935, http://dx.doi.org/10.3390/ijerph10030913
Turner‐Skoff, J. B., & Cavender, N. (2019). The benefits of trees for livable and sustainable communities. Plants, People, Planet. https://doi.org/10.1002/ppp3.39
Kondo, M. C., Fluehr, J. M., McKeon, T., & Branas, C. C. (2018). Urban green space and its impact on human health. International Journal of Environmental Research and Public Health, 15(3). https://doi.org/10.3390/ijerph15030445
Nilsson, K., Bentsen, P., Grahn, P., & Mygind, L. (2019). De quelles preuves scientifiques disposons-nous concernant les effets des forêts et des arbres sur la santé et le bien-être humains? Santé Publique, HS(S1), 219–240. https://www.cairn.info/revue-sante-publique-2019-HS-page-219.htm

I suggest restructuring the manuscript: E.g., large parts of the discussion should be part of the results in terms of a categorisation of previous findings (see my recommendations that I uploaded in the attachment).

In my opinion, more work is needed to make the manuscript self-contained (see my uploaded file).

Experimental design

Please see my comments in the file I uploaded. In my opinion, more work is needed with respect to defining the research question and describing why this knowledge is important.

Validity of the findings

In my opinion, the conclusions could be stated more clearly (please see the file I uploaded).

Additional comments

I like your research. However, in my opinion, a lot more work is needed to make the most out of it (see my comments in the attached file).

---

## Round 0.2 · Minor Revisions

Thank you that you have improved your manuscript. The other reviewer (#2) still wants you to make few minor improvements for your manuscript. These are mainly typological in nature. I see that the language should be better if you consider these suggestions.

Reviewer 1 ·

Basic reporting

The structure and language have been improved by the changes made. Basic reporting is OK as such.

Experimental design

Questions concerning the data and methods have been clarified and are now presented in an acceptable manner.

Validity of the findings

The discussion relates the study to previous research and gives depth to the findings. Conclusions are clear and such that can be drawn from the results.

Additional comments

I find the article greatly improved. Well done!

·

Basic reporting

• Line 77: „complaints“ rather than „complains“?
• Your introduction is very well written, comprehensive, detailed, and unusually long. So, the description of the aim of your study is placed late in the introduction. I suggest starting your long introduction with the purpose of being so detailed about reporting previous findings and contradictory results. In which way are all these previous findings important to understand your study? An alternative may be to move your long literature review from the introduction to the results part (and label it as literature review or literature summary). Subsections 1.2, 1.3, 1.4, 1.6 may suit well in the results part, as you use them to answer your research question, defined in “1.5 Aim of the Study”.
• There seems to be a mistake in subsection numbering in the introduction: “1.5 Aim of the Study” is placed after “1.6. Differences between contemporary and previous generations”.
• The sentence in line 208 (“City children do not have ideal rates.”): Would “City children do not have ideal rates of physical activity” be more precise?
• Line 240: “all of them” rather than “all them”?
• Lines 341 and 342 (“relationship”): Isn’t it differences (between urban and rural parents) that you investigate rather than “relationships”?
• Line 348: I would prefer “the R Commander (Rcmdr) package (Version 2.6.0; Fox, 2005)” instead of “the R Commander add-on”. The full reference is: Fox, J. (2005) The R Commander: A Basic Statistics Graphical User Interface to R. Journal of Statistical Software,14(9): 1–42.
• Line 479: “on the verge of a significant difference” rather than “on the verge of finding a correlation”? Another alternative would be: “…which means that the result represents a nearly significant difference”.
• Lines 480-481: “were found” instead of “we found”.
• Lines 490-491: I suggest “This indicates a statistically significant difference between urban and rural parents” instead of “This indicates a statistical relationship between the results and the place of residence.” It may be confusing for the reader when you switch between “relationship” and “difference” while meaning the same thing. I would recommend consistently talking about “differences” here.
• Line 489: “stronger disagreed” instead of “stronger agreed” in this context.
• Line 510-511: I would prefer “In three out of five categories of tree benefits and disservices only one statement was evaluated differently by urban and rural parents” instead of “In three out of five categories of tree benefits and disservices there was only one statement with statistically significant difference in relation to respondents’ place of living.”
• Line 523: I would not use “their peers” here, because you are talking about children of a previous generation. I would suggest: “…than did children who were born a generation earlier”.
• Line 526: Isn’t it “attitude towards trees” rather than “approach towards trees”?
• Line 527: “agreed to” rather than “agreed with”?
• Line 533: “attitudes” instead of “attitude”?
• Line 533: “higher appreciation” rather than “more appreciation”?
• Line 545: I suggest “…the benefits that nature provides…” instead of “…the benefits that contact with nature contributes to…”.
• Line 569: What do you mean by “gender criteria”? Isn’t it just “gender”? You could also formulate the sentence as follows: “Future research may select a sample which is representative with regard to gender.”
• What do you want to say with the sentence in lines 574-576 “whether the valid theories about the advantages of trees in the environment will be reflected in numerous scientific articles more widely”? This part of the sentence is unclear to me. There are many scientific articles on this topic (and they strengthen theories). What do you mean by “more widely”?
• Line 576: Is there a better word for “space organization”?

Experimental design

no comment

Validity of the findings

no comment

Additional comments

no comment

---

## Round 0.3 · accepted · Accept

Thank you for your time to improve this manuscript. You have well considered suggestions and comments provided by two reviewer during this peer-review process. It is my pleasure to inform that your manuscript is now accepted for the PeerJ.